# Automatic Clipping: Differentially Private Deep Learning Made Easier and Stronger

## Abstract

Per-example gradient clipping is a key algorithmic step that enables practical differential private (DP) training for deep learning models. The choice of clipping threshold $R$, however, is shown to be vital for achieving high accuracy under DP. We propose an easy-to-use replacement, called automatic clipping, that eliminates the need to tune $R$ for any DP optimizers, including DP-SGD, DP-Adam, DP-LAMB and many others. The automatic variants are as private and computationally efficient as existing DP optimizers, but require no DP-specific hyperparameters and thus make DP training as amenable as the standard non-private training. We give a rigorous convergence analysis of automatic DP-SGD in the non-convex setting, which shows that it can enjoy an asymptotic convergence rate that matches the standard SGD, under a symmetric gradient noise assumption of the per-sample gradients. We also demonstrate on various language and vision tasks that automatic clipping outperforms or matches the state-of-the-art, and can be easily employed with minimal changes to existing codebases.

## 1 Introduction

Deep learning has achieved impressive progress in a wide range of tasks. These successes are made available, in part, by the collection of large datasets, sometimes containing sensitive private information of individual data points (e.g., chest scan images, DNA sequences). Prior works have illustrated that deep learning models pose severe privacy risks to individual subjects in the training data and are susceptible to various practical attacks. For example, machine learning services such as Google Prediction API and Amazon Machine Learning can leak membership information from the purchase records (Shokri et al., 2017); if one feeds the GPT2 language model with some specific prefix, the model will autocomplete texts that contain someone's full name, phone number, email address, etc., from the training data that it memorizes (Carlini et al., 2021).

Differential privacy (DP) (Dwork, 2008; Dwork et al., 2014; 2006) is a formal definition of privacy that has been shown to prevent the aforementioned privacy risks in deep learning (Abadi et al., 2016). On a high level, the key difference between the DP deep learning and the regular one is whether the gradient is privately released. In other words, while the standard optimizers update on the summed gradient $\sum_i \boldsymbol{g}_i$, and DP optimizers update on the *private gradient*:

$$\text{DP Optimizer}(\{\boldsymbol{g}_i\}_{i=1}^B) = \text{Optimizer}(\overbrace{\sum_i \boldsymbol{g}_i \cdot \texttt{Clip}(\|\boldsymbol{g}_i\|; R) + \sigma R \cdot \mathcal{N}(0, \mathbf{I})}^{\text{private gradient}}) \quad (1.1)$$

$$\text{Standard Optimizer}(\{\boldsymbol{g}_i\}_{i=1}^B) = \text{Optimizer}(\sum_i \boldsymbol{g}_i) \quad (1.2)$$

Here $\boldsymbol{g}_i \in \mathbb{R}^d$ is the per-sample gradient of loss $l_i$, $\mathcal{N}$ is the standard normal, $\sigma$ is the noise multiplier, and $R$ is the clipping threshold. The clipping function $\texttt{Clip} : \mathbb{R}^d \to \mathbb{R}$ is defined such that $\|\boldsymbol{g}_i \cdot \texttt{Clip}(\boldsymbol{g}_i; R)\| \leq R$. For instance, the DP-SGD in Abadi et al. (2016) on batch $B_t$ is

$$\text{DP-SGD}_{\text{Abadi}} : \quad \boldsymbol{w}_{t+1} = \boldsymbol{w}_t - \eta \Big( \sum_{i \in B_t} \frac{\partial l_i}{\partial \boldsymbol{w}_t} \min \Big( R / \Big\| \frac{\partial l_i}{\partial \boldsymbol{w}_t} \Big\|, 1 \Big) + \sigma R \cdot \mathcal{N}(0, \mathbf{I}) \Big) \quad (1.3)$$

In comparison to the regular training (1.2), two additional DP-specific hyperparameters $R$ and $\sigma$ need to be determined in DP learning (1.1). On the one hand, setting the noise multiplier $\sigma$ is easy and can be derived analytically prior to the training. Whenever the privacy budget $(\epsilon, \delta)$ is determined, one can apply off-the-shelf privacy accounting tools in Section 2.1 to determine $\sigma$, based on the subsampling probability $p$ and the number of iterations $T$:

$$\text{privacy\_accountant}(\sigma, p, T; \delta) = \epsilon$$

On the other hand, the choice of clipping threshold $R$ is crucial to the performance of DP models, yet the hyperparameter tuning is much labor-intensive. Recent advances of DP deep learning on ImageNet (Kurakin et al., 2022) and on E2E datasets (Li et al., 2021), using ResNet18 and GPT2 respectively, illustrate that the performance is very sensitive to $R$. We have reproduced their results in Figure 1. Observe that on ImageNet, ResNet18 can drop from the highest 45% accuracy to 31% if $R$ is chosen 2 times larger, and to 0.1% if $R$ is chosen 4 times larger. Similar drastic drop can also be observed in (Kurakin et al., 2022, Figure 3) even if the noise multiplier $\sigma = 0$. Unlike the noise multiplier $\sigma$, the clipping threshold $R$ cannot be inferred from the privacy budget $(\epsilon, \delta)$ and have to be tuned. Consequently, DP training necessarily requires a 2D grid search for $(R, \eta)$, like the lower plot of Figure 1, whereas the regular training only requires an easy 1D grid search for $\eta$. Even worse, the difficulty of tuning a per-layer clipping threshold vector (McMahan et al., 2018), i.e. one clipping threshold for one layer, may increase exponentially as the number of layers increases.

To save the effort of tuning $R$, previous researches have proposed different approaches. In (Andrew et al., 2021; Pichapati et al., 2019; Golatkar et al., 2022), researchers advocate to use data-adaptive information to select $R$, such as a specified quantile of the gradient norm distribution. These adaptive clipping methods can be a little ad-hoc: they often replace the the need to tune $R$ by the need to tune one or more new hyperparameters, e.g. the quantile to use and the ratio to split the privacy budget between the quantile decision and the gradient perturbation. Another approach used by the practitioners is to replace an expensive 2D grid search by multiple cheaper 1D grid searches. For example, the researchers propose, in (Kurakin et al., 2022, Section 3.3) to fine-tune $\eta$ with non-DP SGD, fix $\eta$ and sweep over various values of the clipping threshold $R$ with DP-SGD, then further fix $R$ and do one more grid search on $\eta$. However, tuning $R$ formally in a data-dependent way (e.g. through cross-validation) introduces additional privacy loss (Papernot & Steinke, 2021), and most existing empirical work does not privately conduct hyperparameter tuning.

We take a completely different route by proposing a new clipping principle that removes $R$, instead of coming up with methods to find the appropriate $R$. We term our method as *automatic clipping* and we term the versions of DP optimizers using it as *automatic DP optimizers*.

We summarize our contributions as follows.

1. We propose the automatic clipping in (4.1) that expunges the clipping threshold from general DP optimizers, allowing DP learning to be as amenable as regular learning.

2. We show that automatic DP optimizers are as private and efficient as existing DP optimizers.

3. We show in Theorem 4 that automatic DP-SGD converges in the non-convex setting, at the same asymptotic convergence rate as the standard SGD. Our theoretical analysis successfully explains the training behaviors in previous empirical works.

4. We demonstrate the superiority of automatic clipping on a variety of vision and language tasks, especially with large models including ResNet, RoBERTa and GPT2.

5. In Appendix K, we include simple code snippets that demonstrate how easy it is to switch from Abadi's clipping to our automatic clipping in popular codebases, e.g. Opacus and ObJAX.

## 2 PRELIMINARIES

### 2.1 DIFFERENTIAL PRIVACY

We consider the $(\epsilon, \delta)$-DP in Definition 2.1, where smaller $(\epsilon, \delta)$ means stronger privacy guarantee.

**Definition 2.1** ((Dwork et al., 2006))**.** A randomized algorithm $M$ is $(\varepsilon, \delta)$-differentially private (DP) if for any two neighboring[1] datasets $S, S'$, and for any event $E$,

$$\mathbb{P}[M(S) \in E] \leqslant e^{\varepsilon} \mathbb{P}[M(S') \in E] + \delta. \qquad (2.1)$$

In words, DP restricts the influence of an arbitrary sample, so that the information contributed by such sample is limited and less vulnerable to privacy attacks. In deep learning, DP is achieved by applying the *subsampled Gaussian mechanism* to privatize the minibatch gradients during training.

As illustrated in Equation (1.1), the subsampled Gaussian mechanism involves (1) Sampling a minibatch by including each data point iid with probability $p$ (2) per-sample gradient clipping to bound

---

[1] $S'$ is a neighbor of $S$ if one can obtain $S'$ by adding or removing one data point from $S$.

the $l_2$ norm sensitivity at $R$ and (3) adding independent Gaussian noise proportional to the sensitivity $R$ and $\sigma$, which is derived from the privacy loss $\epsilon$. This can be realized by leveraging a variety of modern privacy accounting tools, such as Renyi DP (or moments accountant) (Abadi et al., 2016; Mironov, 2017; Wang et al., 2019), Privacy Loss distribution (Fourier accountants) (Koskela et al., 2020; Gopi et al., 2021; Zhu et al., 2022), or Gaussian DP (Dong et al., 2022; Bu et al., 2020).

## 2.2 DIFFERENTIALLY PRIVATE OPTIMIZERS WITH GENERAL CLIPPING OPERATIONS

Privately released stochastic gradients (through the Gaussian mechanism) can be used to instantiate various off-the-shelf optimizers, which gives rise to DP-SGD in (1.3), DP-HeavyBall, DP-AdaGrad, DP-Adam, DP-FedAvg, DP-FedSGD (McMahan et al., 2018), etc. To improve the performance of DP optimizers, previous researches can be classified into two categories.

The first category, where the majority of researches lie in, works with Abadi's clipping and focuses on better design of $R$. To name a few examples, one can adaptively design $R_t$ for each iteration $t$ (Andrew et al., 2021; Pichapati et al., 2019; Golatkar et al., 2022), or design the per-layer clipping threshold vector $\boldsymbol{R} \in \mathbb{R}^L$ for $L$ layers (Abadi et al., 2016; McMahan et al., 2018) so as to apply a different clipping threshold for each layer.

Fewer works fall into the second category that proposes new clipping methods. In fact, any function $\texttt{Clip} : \mathbb{R}^d \to \mathbb{R}$ satisfying $\|\texttt{Clip}(\boldsymbol{g}) \cdot \boldsymbol{g}\| \leq R$ can serve as a valid clipping function besides Abadi's. For instance, the global clipping (Bu et al., 2021b) proposes $\texttt{Clip}_{\text{global}}(\boldsymbol{g}) := \mathbb{I}(\|\boldsymbol{g}\| < R)$ to mitigate the bias of the private gradient and alleviate the mis-calibration issue of DP classifiers. Our automatic clipping also belongs to this category. We note that different clipping methods work orthogonally to optimizers, network architectures and gradient norm computation (see Section 7).

## 3 MOTIVATION

### 3.1 SMALL CLIPPING THRESHOLD WORKS BEST

One intriguing observation that we can make about the recent studies on DP learning with large models is that the state-of-the-art (SOTA) results are often achieved with very small clipping threshold $R$. This observation is consistent in both vision and language tasks. In Li et al. (2021), GPT2 (about 800 million parameters) and RoBERTa models (over 300 millions parameters) achieve the best results under DP on QNLI, MNLI, SST-2, QQP, E2E, and DART datasets, with each per-sample gradient clipped to length $R = 0.1$. In (Kurakin et al., 2022; De et al., 2022; Mehta et al., 2022), ResNets and Vision Transformers achieve the best DP results on ImageNet with $R = 1$; in (Tramer & Boneh, 2020), the best DP results on CIFAR10 use $R = 0.1$ with ResNeXt-29 and SimCLRv2 (Chen et al., 2020a). The effectiveness of small clipping threshold together with proper learning rate is depicted in Figure 1.

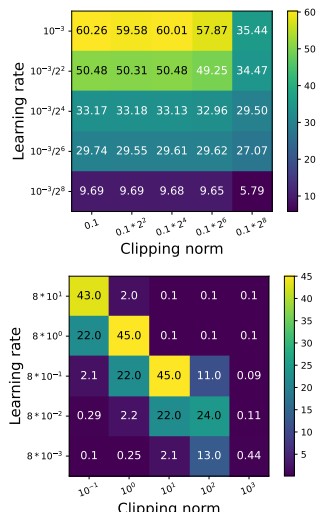

Figure 1: Ablation study of clipping threshold and learning rate. Upper: BLEU score of GPT2 on E2E dataset (Li et al., 2021), with DP-AdamW. Lower: Test accuracy of ResNet18 on ImageNet dataset (Kurakin et al., 2022), with DP-SGD and momentum.

Intuitively, smaller $R$ implies that the Abadi's clipping (3.1) happens, which means $\min\left(R/\|\boldsymbol{g}_i\|, 1\right) = R/\|\boldsymbol{g}_i\|$. Given that the clipping threshold $R$ is so small compared to the number of parameters in large neural networks, and that strong DP is guaranteed when the number of training iterations is small (i.e. $\|\boldsymbol{g}_i\|$ has not converged to small values yet), we expect and empirically observe that the clipping happens on a large proportion of per-sample gradients at all iterations. For instance, we find in the GPT2 generation experiments in Li et al. (2021) that 100% of per-sample gradients are clipped at all iterations; in classification tasks such as QQP/QNLI/MNLI, the percentage of clipping is about $20 \sim 60\%$ on average (more details in Appendix H.1).

### 3.2 PER-SAMPLE GRADIENT NORMALIZATION AS NEW CLIPPING

In the small clipping threshold regime, we can approximately view

$$\texttt{Clip}_{\text{Abadi}}(\boldsymbol{g}_i; R) = \min\left(R/\|\boldsymbol{g}_i\|, 1\right) \approx R/\|\boldsymbol{g}_i\| =: \texttt{Clip}_{\text{AUTO-V}}(\boldsymbol{g}_i; R) \tag{3.1}$$

and thus derive a novel private gradient $\sum_i R \frac{\boldsymbol{g}_i}{\|\boldsymbol{g}_i\|} + \sigma R \cdot \mathcal{N}(0, \mathbf{I})$. Here AUTO-V stands for the vanilla automatic clipping, which essentially performs the gradient normalization on each per-sample gradient. As a specific example, we can write the $R$-dependent automatic DP-SGD as

$$R\text{-dependent DP-SGD}_{\text{AUTO-V}} : \boldsymbol{w}_{t+1} = \boldsymbol{w}_t - \eta \Big( \sum_{i \in B_t} R \frac{\partial l_i}{\partial \boldsymbol{w}_t} / \| \frac{\partial l_i}{\partial \boldsymbol{w}_t} \| + \sigma R \cdot \mathcal{N}(0, \mathbf{I}) \Big) \quad (3.2)$$

We may view our AUTO-V clipping as to maximize the dot-product similarity (a commonly used similarity measure, e.g. in the attention block in transformers (Vaswani et al., 2017)) between the clipped gradient and the regular gradient. Suppose we want

$$\max_{C_i} \left\langle \sum_i C_i \boldsymbol{g}_i, \sum_j \boldsymbol{g}_j \right\rangle \quad \text{s.t. } 0 \le C_i \le R/\|\boldsymbol{g}_i\|$$

Note that the constraint is a sufficient condition for clipping, as discussed in Section 2.2. It is not hard to see that the optimal clipping factor is

$$C_i = R/\|\boldsymbol{g}_i\| \cdot \mathbb{I}(\langle \boldsymbol{g}_i, \sum_j \boldsymbol{g}_j \rangle > 0)$$

If the per-sample gradients are indeed concentrated in the sense $\forall i, \langle \boldsymbol{g}_i, \sum_j \boldsymbol{g}_j \rangle \ge 0$, then AUTO-V is the optimal per-sample gradient clipping. We compare with Abadi's clipping in Figure 2, where the dot-product similarity is significantly magnified by our AUTO-V clipping.

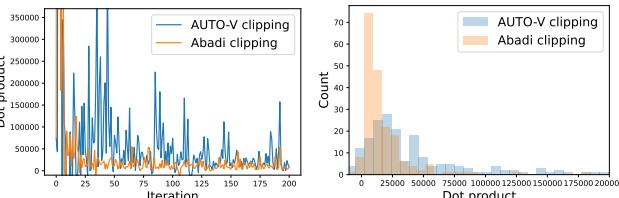

Figure 2: RoBERTa-base with DP-Adam ($\epsilon = 3$) on SST2 dataset, as in Section 6.2.

### 3.3 STABILITY CONSTANT BREAKS SCALE-INVARIANCE AND REMAINS STATIONARY

One potential drawback of AUTO-V clipping is that all gradients lose their magnitudes information completely, since $\|\boldsymbol{g}_i \cdot \text{Clip}_{\text{AUTO-V}}(\boldsymbol{g}_i; R)\| = R, \forall i$. This scale-invariance in AUTO-V and partially in Abadi's clipping (when $\|\boldsymbol{g}_i\| > R$) leads to the "lazy region" issue: the parameters will not be updated by DP-GD even if the true gradients are non-zero. In Figure 3, we illustrate in a logistic regression[2] that AUTO-V and Abadi's clipping have zero clipped gradient for the trainable parameter $\theta \in [-2, 2]$, as the per-sample gradients from two classes cancel each other.

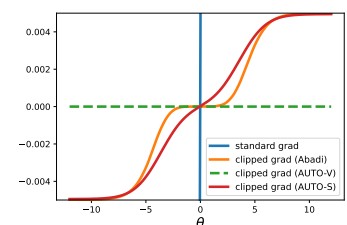

Figure 3: Gradient (scalar) at each $\theta$.

Another benefit of $\gamma$ is to remain stationary as $\boldsymbol{g}_i \to 0$, i.e. making the clipped gradient $C_i \boldsymbol{g}_i \to \boldsymbol{g}_i / \gamma$ small rather than having a magnitude $R$ in AUTO-V. We elaborate this point in Section 4.3.

To preserve the magnitude information and thus escape the lazy region, we propose the AUTO-S clipping, with a positive stability constant $\gamma$:

$$\text{Clip}_{\text{AUTO-S}}(\boldsymbol{g}_i; R) := R/(\|\boldsymbol{g}_i\| + \gamma) \quad (3.3)$$

We visualize in Figure 4 that AUTO-S allows larger per-sample gradients to have larger magnitudes after the clipping, while still allowing smaller gradients to vanish after "clipping". This is critical in our convergence analysis and allows DP-SGD$_{\text{AUTO-S}}$ (but not DP-SGD$_{\text{AUTO-V}}$) to converge to zero gradient norms in Section 5.

## 4 AUTOMATIC DP TRAINING

One may wonder why our clipping (3.1)(3.3) is automatic at all, if the hyperparameter $R$ is still present and there is an additional parameter $\gamma$ to choose. It turns out that any constant choice of $R > 0$ is equivalent to choosing $R = 1$, and common deep learning optimizers are insensitive

---

[2]The settings are in Appendix F, where the lazy region issues also emerge in the mean estimation problem. We note that the lazy region is also discussed in (Chen et al., 2020b, Example 2).

to the choice of $\gamma$ (e.g. for any $\gamma > 0$, we show that the gradient norm converges to zero at the same asymptotic rate in Theorem 4; see also the ablation study in Figure 14). Consequently, we set $\gamma = 0.01$ as the default. Specifically, let us redefine the $R$-independent clipping function:

$$\texttt{Clip}_{\text{AUTO-S}}(\boldsymbol{g}_i) := 1/(||\boldsymbol{g}_i|| + \gamma). \tag{4.1}$$

With this clipping, we can design automatic DP optimizers similar to (1.1):

$$\text{Automatic DP Optimizer}(\{\boldsymbol{g}_i\}_{i=1}^{B}) = \text{Optimizer}\Big(\underbrace{\sum_{i \in B_t} \frac{\boldsymbol{g}_{t,i}}{||\boldsymbol{g}_{t,i}|| + \gamma} + \sigma \cdot \mathcal{N}(0, \mathbf{I})}_{\text{denoted as } \hat{\boldsymbol{g}}_t}\Big) \tag{4.2}$$

Clearly, the new private gradient $\hat{\boldsymbol{g}}_t$ from our automatic clipping is $R$-independent, in contrast to the one used in (1.1). A concrete example (in the case of $\gamma = 0$) that is comparable to (3.2) will be

$$R\text{-independent DP-SGD}_{\text{AUTO-V}}: \quad \boldsymbol{w}_{t+1} = \boldsymbol{w}_t - \eta\Big(\sum_{i \in B_t} \frac{\partial l_i}{\partial \boldsymbol{w}_t}\Big/\Big\|\frac{\partial l_i}{\partial \boldsymbol{w}_t}\Big\| + \sigma \cdot \mathcal{N}(0, \mathbf{I})\Big) \tag{4.3}$$

Leveraging the private gradient $\hat{\boldsymbol{g}}_t$ in (4.2), we can train DP neural networks without tuning DP-specific hyperparamters $R$ and $\sigma$, as demonstrated in Algorithm 1.

---

**Algorithm 1** Automatic Deep Learning with DP

---

**Parameters:** initial weights $\boldsymbol{w}_0$, learning rate $\eta_t$, sampling probability $p$, number of iterations $T$.

1: Find $\sigma$ such that $\epsilon_{\text{Accountant}}(\delta, \sigma, p, T) \leq \epsilon$ from any privacy accountant.
2: **for** iteration $t = 1, \cdots, T$ **do**
3:     Sample a batch $B_t$ by including each data point i.i.d. with probability $p$
4:     Apply automatic clipping to per-sample gradients $\{\boldsymbol{g}_i\}_{i \in B_t}$: $\hat{\boldsymbol{g}}_i = \boldsymbol{g}_i/(||\boldsymbol{g}_i||_2 + 0.01)$.
5:     Add Gaussian noise to the sum of clipped gradients: $\hat{\boldsymbol{g}} = \sum_i \hat{\boldsymbol{g}}_i + \sigma \cdot \mathcal{N}(0, \mathbf{I})$.
6:     Update $\boldsymbol{w}_t$ by any optimizer on the private gradient $\hat{\boldsymbol{g}}$ with learning rate $\eta_t$.

---

We will elaborate two distinct reasons in the next sub-sections for the following statement:

> DP Optimizer$_{\text{Abadi}}$ $\approx$ $R$-dependent DP Optimizer$_{\text{AUTO}}$ $\equiv$ $R$-independent DP Optimizer$_{\text{AUTO}}$

which reduces the hyperparameter tuning of DP training to that of the regular training, i.e. only on learning rate, weight decay, etc. The significant save in the tuning effort is illustrated in Figure 15.

## 4.1 Non-adaptive optimizer couples clipping threshold with learning rate

With $R$-dependent automatic clipping, DP-SGD becomes

$$\boldsymbol{w}_{t+1} = \boldsymbol{w}_t - \eta\Big(\sum_{i \in B_t} \boldsymbol{g}_{t,i} \cdot \frac{R}{||\boldsymbol{g}_{t,i}|| + \gamma} + \sigma R \cdot \mathcal{N}(0, \mathbf{I})\Big) = \boldsymbol{w}_t - \eta R \hat{\boldsymbol{g}}_t.$$

We can view $\eta_{\text{effective}} \equiv \eta R$ as a whole: increasing $R$ has the same effect as increasing $\eta$, which explains the diagonal pattern in Figure 1(lower plot) where DP-SGD$_{\text{Abadi}}$ is applied with small clipping threshold[3]. We extend to general non-adaptive optimizers in Theorem 1 [4].

**Theorem 1.** *Non-adaptive $R$-dependent automatic DP optimizers (including SGD, Heavy-ball(Polyak, 1964) and NAG(Nesterov, 1983)), with learning rate $\eta$ and weight decay $\lambda$, is equivalent to $R$-independent automatic DP optimizers, with learning rate $\eta R$ and weight decay $\lambda/R$.*

## 4.2 Adaptive optimizer can be insensitive to clipping threshold

Adaptive automatic DP optimizers are different than the non-adaptive ones, as the clipping threshold cancels out instead of being coupled with learning rate. To see this, we scrutinize DP-Adam$_{\text{Abadi}}$ (which is similar to DP-Adam$_{\text{AUTO-V}}$) in Figure 1(upper plot), where columns to the left are almost identical. Further evidence is observed in (Mehta et al., 2022, Table 5) that shrinking $R$ has zero effect on LAMB. We now give a simple explanation using AdaGrad (Duchi et al., 2011):

$$\boldsymbol{w}_{t+1} = \boldsymbol{w}_t - \eta\frac{\boldsymbol{g}_t}{\sqrt{G_t}}$$

---

[3]When we further consider weight decay in automatic clipping (included in Theorem 1), increasing $R$ is no longer equivalent to increasing $\eta$, as $\eta$ also couples with the weight decay constant $\lambda$.

[4]This coupling of $\eta$ and $R$ is also partially observed in (De et al., 2022) through a reparameterization trick of Abadi's clipping. Unlike AUTO-S/V, their coupling is not strict (e.g. doubling $R$ is not equivalent to doubling $\eta$ in their Figure 8, thus necessitating tuning both $(\eta, R)$), and the relationship to weight decay was not discussed.

where $\boldsymbol{g}_t = \sum_i \boldsymbol{g}_{t,i}$ is the gradient sum and $G_t = \sum_{\tau < t} \boldsymbol{g}_\tau^2$ is sum of gradient square by Hadamard product over the past iterations. In $R$-dependent DP-AdaGrad$_{\text{AUTO-V}}$, the private gradient is $R\hat{\boldsymbol{g}}_t$ in place of the standard gradient sum $\boldsymbol{g}_t$, and $\hat{G}_t = R^2 \sum_{\tau \le t} \hat{\boldsymbol{g}}_\tau^2$:

$$\boldsymbol{w}_{t+1} = \boldsymbol{w}_t - \eta \frac{R\hat{\boldsymbol{g}}_t}{\sqrt{\hat{G}_t}} = \boldsymbol{w}_t - \eta \frac{\hat{\boldsymbol{g}}_t}{\sqrt{\sum_{\tau<t}(\hat{\boldsymbol{g}}_\tau)^2}}.$$

We generalize to the general adaptive optimizers in Theorem 2 .

**Theorem 2.** *Adaptive $R$-dependent automatic DP optimizers (e.g. AdaGrad(Duchi et al., 2011), AdaDelta(Zeiler, 2012), AdaMax/Adam(Kingma & Ba, 2014), NAdam(Dozat, 2016), RAdam(Liu et al., 2019a), LARS(You et al., 2017), LAMB(You et al., 2020)), with learning rate $\eta$ and weight decay $\lambda$ is equivalent to $R$-independent automatic DP optimizers with learning rate $\eta$ and weight decay $\lambda/R$. With decoupled weight decay(Loshchilov & Hutter, 2018), $R$-dependent automatic DP-AdamW is equivalent to $R$-independent automatic DP-AdamW with the same $\eta$ and $\lambda$.*

In Appendix B.3, we also analyze the automatic DP optimizers with per-layer clipping style.

### 4.3 AUTOMATIC CLIPPING IS EQUALLY PRIVATE AND MAXIMIZES UTILITY

In Theorem 3 (proved in Appendix A), we show that the new private gradient $\hat{\boldsymbol{g}}_t$ in (4.2) has the same level of privacy guarantee as the existing one in (1.1), since the global sensitivity remains the same (see Figure 4). We note that as long as $\gamma > 0$, the magnitude information of per-sample gradients is preserved by AUTO-S, in the sense that $\|\boldsymbol{g}_i\| > \|\boldsymbol{g}_j\| \iff \|C_i\boldsymbol{g}_i\| > \|C_j\boldsymbol{g}_j\|$, whereas this can be violated in both the AUTO-V and Abadi's clipping (as depicted by the flat curve in Figure 4 when $\|\boldsymbol{g}_i\| > 1$).

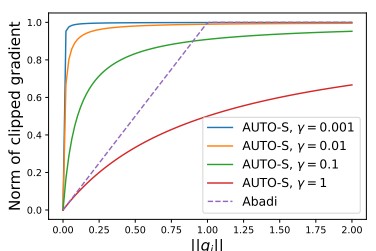

Figure 4: Gradient norms before and after clipping by different methods at $R = 1$.

Additionally, note that when $\gamma$ is small, almost all data points "max out" the signal relative to the amount of noise we add. To say it differently, for the same amount of noise, AUTO-S with small $\gamma$ allows more signal to be pushed through a differentially private channel. Towards the end of the training, i.e., at the limit when $\|\boldsymbol{g}_i\| \to 0$ for all $i$, then we have $\sum_i \frac{\boldsymbol{g}_i}{\|\boldsymbol{g}_i\|+\gamma} \to \frac{1}{\gamma} \sum_i \boldsymbol{g}_i$. In words, the clipped gradients become closer to the standard SGD, thus do not suffer from the instability of AUTO-V.

**Theorem 3.** *Under the noise multiplier $\sigma$, number of iterations $T$, subsampling probability $B/n$, DP optimizers using AUTO-V or AUTO-S clipping satisfy $(\epsilon_{\text{Accountant}}(\delta, \sigma, B/n, T), \delta)$-DP, where $\epsilon_{\text{Accountant}}$ is any valid privacy accountant for DP-SGD under Abadi's clipping.*

## 5 CONVERGENCE ANALYSIS OF DP-SGD WITH AUTOMATIC CLIPPING

### 5.1 CONVERGENCE THEORY OF DP-SGD TO STATIONARY POINTS

We highlight that automatic clipping can be more amenable to analysis than Abadi's clipping in Chen et al. (2020b), since we no longer need to decide whether each per-sample gradient is clipped.

To analyze the convergence of automatic DP-SGD (4.2) in the non-convex setting, we follow the standard assumptions in the SGD literature (Ghadimi & Lan, 2013; Allen-Zhu, 2018; Bottou et al., 2018), including a symmetry assumption on the gradient noise, which is empirically verified in (Chen et al., 2020b, Figure 3)[5].

**Assumption 5.1** (Lower bound of loss). For all $\boldsymbol{w}$ and some constant $\mathcal{L}_*$, we have $\mathcal{L}(\boldsymbol{w}) \ge \mathcal{L}_*$.

**Assumption 5.2** (Smoothness). Let $\boldsymbol{g}(\boldsymbol{w})$ denote the gradient of the objective $\mathcal{L}(\boldsymbol{w})$. Then $\forall \boldsymbol{w}, \boldsymbol{v}$, there is an non-negative constant $L$ such that

$$\mathcal{L}(\boldsymbol{v}) - \left[\mathcal{L}(\boldsymbol{w}) + \boldsymbol{g}(\boldsymbol{w})^\top(\boldsymbol{v} - \boldsymbol{w})\right] \le \frac{L}{2}\|\boldsymbol{w} - \boldsymbol{v}\|^2. \tag{5.1}$$

---

[5]This symmetry assumption is relaxed from the Gaussian noise assumption (since a zero-mean Gaussian is symmetric) in the SGD literature (Mandt et al., 2017; Smith et al., 2018; Chaudhari & Soatto, 2018; Xie et al., 2020). By setting minibatch size to 1, we reduce the noise assumption to a per-sample gradient case.

**Assumption 5.3** (Gradient noise). The per-sample gradient noise $\tilde{\boldsymbol{g}}_{t,i} - \boldsymbol{g}_t$ is i.i.d. from some ditribution such that

$$\mathbb{E}(\tilde{\boldsymbol{g}}_{t,i} - \boldsymbol{g}_t) = 0, \mathbb{E}\|\tilde{\boldsymbol{g}}_{t,i} - \boldsymbol{g}_t\|^2 \leq \xi^2,$$

and $\tilde{\boldsymbol{g}}_{t,i}$ is centrally symmetric about $\boldsymbol{g}_t$ in distribution: $\tilde{\boldsymbol{g}}_{t,i} - \boldsymbol{g}_t \overset{\mathcal{D}}{=} \boldsymbol{g}_t - \tilde{\boldsymbol{g}}_{t,i}$.

We show in Theorem 4 that DP-SGD with AUTO-S clipping allows the true gradient norm to converge to zero, though the clipped gradient may still be biased, but not so with AUTO-V clipping. We leave the proof in Appendix C.1.

**Theorem 4.** *Under Assumption 5.1, 5.2, 5.3, running DP-SGD with automatic clipping for $T$ iterations and setting the learning rate $\eta \propto 1/\sqrt{T}$ give*[6]

$$\min_{0 \leq t \leq T} \mathbb{E}(\|\boldsymbol{g}_t\|) \leq \mathcal{G}\left(\frac{4}{\sqrt{T}}\sqrt{(\mathcal{L}_0 - \mathcal{L}_*)L\left(1 + \frac{\sigma^2 d}{B^2}\right)}; \xi, \gamma\right) := \min_{r > 0}\frac{\xi}{r} + \mathcal{F}(\cdots; r, \xi, \gamma). \quad (5.2)$$

*Here $\cdots$ represents the first argument of $\mathcal{G}$, and $\mathcal{G}$ is increasing and positive. As $T \to \infty$, we have $\min_t \mathbb{E}(\|\boldsymbol{g}_t\|) = O(T^{-1/4})$ for AUTO-S, the same rate as the standard SGD given in Theorem 9.*

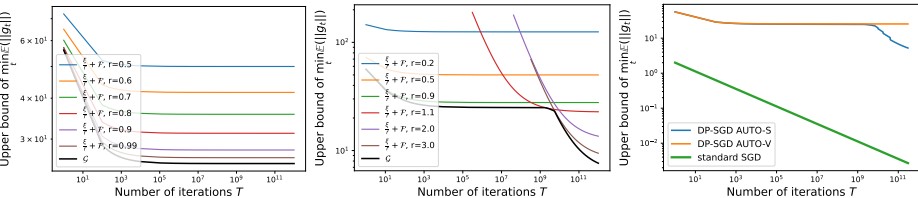

Figure 5: Left: DP-SGD with AUTO-V clipping. Middle: DP-SGD with AUTO-S clipping. Right: Log-log plot of convergence rate in comparison to standard SGD. Here $\xi = 25, \gamma = 0.01$, and the $O(1/\sqrt{T})$ term is set to 10 for DP-SGD and to 2 for standard SGD.

**Remark 5.4.** We show in Theorem 6 and in Figure 5 that the upper bound (5.2) has $\mathcal{G} \geq \xi$ for AUTO-V ($\gamma = 0$), and $\mathcal{G}$ only reduces to zero for AUTO-S ($\gamma > 0$). We provide real data evidence in Figure 13 that strictly positive $\gamma$ reduces the gradient norm significantly.

## 5.2 ANALYSIS OF FACTORS AFFECTING THE CONVERGENCE

We now analyze the many factors that affect the convergence in Theorem 4, from a unified viewpoint of both the convergence and the privacy.

We start with the stability constant $\gamma$ and the learning rate $\eta_t$, both only affect the convergence not the privacy. We empirically observe in Figure 7 that small $\gamma$ benefits the convergence at initial iterations (when the privacy guarantee is strong) but larger $\gamma$ converges faster asymptotically. For $\eta_t$, the optimal is in fact the miminizer of the hyperbola in (C.4), that is unique and tunable.

Next, we focus on the hyperparameters that affect both convergence and privacy: the batch size $B$, the noise multiplier $\sigma$, and the number of iterations $T$. These hyperparameters have to be considered along the privacy-accuracy tradeoff, not just from a convergence perspective.

Recall that given a fixed privacy budget $(\epsilon, \delta)$, we rely on modern privacy accountant for computing the appropriate combinations of parameter $\sigma, T, B$. The exact expression of the bound as a function of $(\epsilon, \delta)$ is somewhat messy. For this reason, we illustrate our analysis in terms of the surrogate parameter $\mu$ for $\mu$-GDP (Dong et al., 2022). Bu et al. (2020) showed that DP-SGD's privacy guarantee asymptotically converges to $\mu$-GDP (as $T \to \infty$) with $\mu = \frac{B}{n}\sqrt{T(e^{1/\sigma^2} - 1)}$. $\mu$-GDP implies $(\epsilon, \delta)$-DP with $\epsilon = \mu^2 + \mu\sqrt{2\log(1/\delta)})$[7]. We can alternatively leverage $\rho$-tCDP (Bun et al., 2018) for similar conclusions, using $\rho$ in place of $\mu^2$ in (5.3).

---

[6]The upper bound takes an implicit form of $\mathcal{G}(\cdot; \xi, \gamma)$ because it is a lower envelope of functions $\frac{\xi}{r} + \mathcal{F}(\cdot; r, \xi, \gamma)$ over all possible $r > 0$, whose forms are detailed in Theorem 6. Notice that $\mathcal{G}$ results only from the clipping operation, not from the noise addition.

[7]More precisely, $\mu$-GDP is equivalent to an entire family of $(\epsilon, \delta)$-DP for any $\epsilon > 0$ and $\delta = \Phi(\mu/2 - \epsilon/\mu) - e^{\epsilon}\Phi(-\mu/2 - \epsilon/\mu)$ where $\Phi$ is the standard Gaussian CDF.

**Theorem 5.** *Under Assumption 5.1, 5.2, 5.3, fixing the asymptotic $\mu(\epsilon, \delta)$-GDP parameter, running DP-SGD with automatic clipping for $T$ iterations and setting the learning rate $\eta \propto 1/\sqrt{T}$ give*

$$\min_{0 \leq t \leq T} \mathbb{E}(\|\boldsymbol{g}_t\|) \leq \mathcal{G}\left(4\sqrt{(\mathcal{L}_0 - \mathcal{L}_*)L\left(\frac{1}{T} + \frac{d}{\mu^2 n^2} + O(\frac{1}{B^2 T})\right)}; \xi, \gamma\right) \tag{5.3}$$

To show that our analysis matches the training behaviors observed in SOTA empirical work (Li et al., 2021; Kurakin et al., 2022; De et al., 2022; Tramer & Boneh, 2020; Mehta et al., 2022; Yu et al., 2021), we minimize the first argument of $\mathcal{G}$ in (5.3), denoted as $X(B, T, \mu, d, L, \mathcal{L}_0)$.

1. **[Train longer with larger noise]** Fixing the expected batch size $B$, we see that $X$ is decreasing in $T$. Hence larger $T$ and consequently larger $\sigma$ are preferred.

2. **[Larger batch size helps]** Fixing number of iterations $T$ or epochs $E = BT/n$, we see that $X$ is decreasing in $B$. Hence larger $B$ and consequently larger $\sigma$ are preferred.

3. **[Pretraining is critical]** Pretraining can boost the DP accuracy through a much smaller initial loss $\mathcal{L}_0$ and from a smooth (small $L$) and flat (small $\xi$, c.f. Figure 7(left)) initialization.

4. **[Learning rate needs tuning]** The optimal learning rate by minimizing (C.4) is $\sqrt{\frac{(\mathcal{L}_0 - \mathcal{L}_*)\mu^2 n^2}{L(\mu^2 n^2 + dT)}}$. This indicates that one should use larger learning rate for smaller model $d$, weaker privacy (larger $\mu$ or small $\epsilon$), or smaller iteration budget $T$. Interestingly, the optimal choice of learning rate is independent to (expected) batch-size $B$.

## 6 EXPERIMENTS

We evaluate our automatic DP training on image classification, sentence classification, and table-to-text generation tasks. Detailed settings including hyperparameters can be found in Appendix G.

### 6.1 IMAGE CLASSIFICATION

For MNIST/FashionMNIST, we use the same setup as in (Papernot et al., 2021; Tramer & Boneh, 2020; Shamsabadi & Papernot, 2021) with a simple CNN. For CIFAR10, we use the same setup as in Tramer & Boneh (2020) with pretrained SimCLRv2 (Chen et al., 2020a). For ImageNette, a 10-class sub-task of ImageNet (Deng et al., 2009), we use the same setup as in Klause et al. (2022) without the learning rate decay. For CelebA (Liu et al., 2015), the real human face dataset, we train ResNet9 (He et al., 2016) with group normalization to replace the batch normalization. Notice that CelebA contains high-resolution (178x218) images, each with 40 labels. We consider CelebA for either multi-class classification on one label, e.g. 'Smiling' and 'Male', or for multi-label/multi-task problem to learn all labels simultaneously.

| Task | Model | $(\epsilon, \delta)$ | Accuracy % | | non-DP |
|---|---|---|---|---|---|
| | | | Abadi's clipping | AUTO-S clipping | $(\epsilon = \infty)$ |
| MNIST | 4-layer CNN | $(3, 1e\text{-}5)$ | $98.04 \pm 0.09$ | $98.15 \pm 0.07$ | $99.11 \pm 0.07$ |
| FashionMNIST | 4-layer CNN | $(3, 1e\text{-}5)$ | $86.04 \pm 0.26$ | $86.36 \pm 0.18$ | $89.57 \pm 0.13$ |
| CIFAR10 pretrained | SimCLRv2 | $(2, 1e\text{-}5)$ | $92.44 \pm 0.13$ | $92.70 \pm 0.02$ | $94.42 \pm 0.01$ |
| ImageNette | ResNet9 | $(8, 1e\text{-}4)$ | $60.29 \pm 0.53$ | $60.71 \pm 0.48$ | $71.11 \pm 0.37$ |
| CelebA [Smiling] | ResNet9 | $(8, 5e\text{-}6)$ | $90.75 \pm 0.11$ | $91.08 \pm 0.08$ | $92.61 \pm 0.20$ |
| CelebA [Male] | ResNet9 | $(8, 5e\text{-}6)$ | $95.54 \pm 0.14$ | $95.70 \pm 0.07$ | $97.90 \pm 0.04$ |
| CelebA Multi-label | ResNet9 | $(3, 5e\text{-}6)$ | $86.81 \pm 0.03$ | $87.05 \pm 0.01$ | $90.30 \pm 0.02$ |
| CelebA Multi-label | ResNet9 | $(8, 5e\text{-}6)$ | $87.52 \pm 0.15$ | $87.58 \pm 0.04$ | $90.30 \pm 0.02$ |

Table 1: Average test accuracy and 95% confidence interval on image tasks over 5 runs.

In Table 1, we observe that AUTO-S clipping outperforms existing clipping in all datasets with statistical significance. Interestingly, the standard deviation from different runs is smaller for automatic DP optimizers, indicating better reproducibility and stability. We additionally experiment 40 binary classification problems on CelebA with respect to each label, and observe that the mean accuracy further improves to 91.63% at $\epsilon = 8$ for AUTO-S (see Appendix J).

### 6.2 SENTENCE CLASSIFICATION

On five benchmark language datasets (MNLI(m/mm)(Williams et al., 2018), QQP(Iyer et al., 2017), QNLI(Rajpurkar et al., 2016), SST2(Socher et al., 2013)), we compare our automatic DP training with reparameterized gradient perturbation (RGP, (Yu et al., 2021)) and full-parameter finetuning (full, (Li et al., 2021)) using RoBERTa models (Liu et al., 2019b). These methods use the same

experimental setup. For language models, our automatic training is based on the codebase of (Li et al., 2021)[8].

| Method | $\epsilon = 3$ | | | | $\epsilon = 8$ | | | | $\epsilon = \infty$ (non-DP) | | | |
|---|---|---|---|---|---|---|---|---|---|---|---|---|
| | MNLI | QQP | QNLI | SST2 | MNLI | QQP | QNLI | SST2 | MNLI | QQP | QNLI | SST2 |
| RGP (Yu et al., 2021) | - | - | - | - | 80.5/79.6 | 85.5 | 87.2 | 91.6 | 83.6/83.2 | 89.3 | 91.3 | 92.9 |
| full (Li et al., 2021) | 82.45/82.99 | 85.56 | **87.42** | 91.86 | 83.20/83.46 | 86.08 | **87.94** | 92.09 | | | | |
| full AUTO-V | 81.21/82.03 | 84.72 | 86.56 | 91.86 | 82.18/82.64 | **86.23** | 87.24 | 92.09 | 85.91/86.14 | 87.34 | 91.40 | 94.49 |
| full AUTO-S | **83.22/83.21** | 85.76 | 86.91 | **92.32** | **83.82/83.55** | 86.58 | 87.85 | **92.43** | | | | |

Table 2: Test accuracy on language tasks with RoBERTa-base (12 blocks, 125 million parameters).

| Method | $\epsilon = 3$ | | | | $\epsilon = 8$ | | | | $\epsilon = \infty$ (non-DP) | | | |
|---|---|---|---|---|---|---|---|---|---|---|---|---|
| | MNLI | QQP | QNLI | SST2 | MNLI | QQP | QNLI | SST2 | MNLI | QQP | QNLI | SST2 |
| RGP (Yu et al., 2021) | - | - | - | - | 86.1/86.0 | 86.7 | 90.0 | 93.0 | - | - | - | - |
| full (Li et al., 2021) | 86.43/86.46 | 86.43 | 90.76 | 93.04 | 87.02/**87.26** | 87.47 | 91.10 | 93.81 | | | | |
| full AUTO-V | 85.33/85.61 | **86.61** | 89.99 | **93.12** | 85.91/86.10 | 86.86 | 90.55 | 93.35 | 90.33/90.03 | 87.90 | 93.61 | 96.21 |
| full AUTO-S | 86.27/**86.67** | **86.76** | **91.01** | **93.92** | **87.07**/87.16 | 87.47 | **91.45** | **94.61** | | | | |

Table 3: Test accuracy on language tasks with RoBERTa-large (24 blocks, 355 million parameters).

In Table 2 and Table 3, we note that full parameter finetuning with AUTO-S outperforms or at least matches SOTA on all tasks. We use *exactly the same* hyperparameters as in Li et al. (2021).

### 6.3 TABLE-TO-TEXT GENERATION

We compare our automatic DP training with a variety of fine-tuning methods, for table-to-text generation task on E2E dataset (Dusek et al., 2020), where the goal is to generate texts about different aspects of a restaurant's data. We measure the success on this task by BLEU, ROUGE-L (in Table 4), METEOR, NIST, CIDEr (extended in Table 8), with higher value meaning better model quality.

| Metric | DP guarantee | GPT2 large full AUTO-S | GPT2 medium full AUTO-S | GPT2 | | | | | | | |
|---|---|---|---|---|---|---|---|---|---|---|---|
| | | | | full AUTO-S | full AUTO-V | full (Li et al., 2021) | LoRA (Hu et al., 2021) | RGP (Yu et al., 2021) | prefix (Li & Liang, 2021) | top2 | retrain |
| BLEU | $\epsilon = 3$ | **64.180** | **63.850** | **61.340** | 61.519 | 61.519 | 58.153 | 58.482 | 47.772 | 25.920 | 15.457 |
| | $\epsilon = 8$ | **64.640** | **64.220** | **63.600** | 63.189 | 63.189 | **63.389** | 58.455 | 49.263 | 26.885 | 24.247 |
| | non-DP | 66.840 | 68.500 | 69.463 | 69.463 | 69.463 | 69.682 | 68.328 | 68.845 | 65.752 | 65.731 |
| ROGUE-L | $\epsilon = 3$ | **67.857** | **67.071** | **65.872** | 65.670 | 65.670 | **65.773** | 65.560 | 58.964 | 44.536 | 35.240 |
| | $\epsilon = 8$ | **68.968** | **67.533** | 67.073 | 66.429 | 66.429 | **67.525** | 65.030 | 60.730 | 46.421 | 39.951 |
| | non-DP | 70.384 | 71.458 | 71.359 | 71.359 | 71.359 | 71.709 | 68.844 | 70.805 | 68.704 | 68.751 |

Table 4: Test performance on E2E dataset with GPT2. Additional performance measures are included in Table 8. The best two GPT2 models for each row are marked in bold.

Competitive methods include low-rank adaption (LoRA), prefix-tuning (prefix), RGP, only fine-tuning the top 2 Transformer blocks (top2), and training from scratch (retrain), as were recorded in Li et al. (2021). Again, we use the *exactly the same* hyperparameters as in Li et al. (2021). For GPT2 (124 million parameters), GPT2 medium (355 million), and GPT2 large (774 million), Table 4 shows that AUTO-S is scalable with stronger performance on larger models. Our automatic full-parameter finetuning has the best overall performance. Additionally, we highlight that AUTO-S and methods like LoRA are not mutually exclusive and can be combined to yield strong performance, since AUTO-S modifies the optimizers and LoRA modifies the architecture.

## 7 DISCUSSION

In this work, we proposed the automatic clipping as a drop-in replacement to the standard per-example clipping differentially private training. This is the first technique that eliminate the need to tune the clipping threshold $R$, thus making DP deep learning as easy as regular learning. Our AUTO-S method enjoys both theoretical guarantee of convergence in non-convex problems (under various conditions), and strong empirical performance that advances the state-of-the-art (SOTA) of DP learning on both computer vision and language tasks.

We are excited about the future of automatic DP training, especially along with other working techniques. Notably, our automatic clipping applies compatibly with general optimizers (e.g. (Bu et al., 2021a; Du & Mi, 2021)), clipping styles (all-layer or per-layer), architecture modifications (e.g. LoRA, RGP, prefix), and data augmentation (e.g. adversarial training (Goodfellow et al., 2015) and multiple augmentation De et al. (2022)). Thus, we expect to achieve comparable results to all SOTA in a lightweight fashion.

---

[8]See https://github.com/lxuechen/private-transformers and the detailed modification in Appendix K.3.

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
