# OpenReview forum: "Automatic Clipping: Differentially Private Deep Learning Made Easier and Stronger"
_ICLR.cc/2023/Conference — Submitted to ICLR 2023_

### Official Review · Reviewer_6ygh · 2022-10-25

**Confidence:** 3
**Correctness:** 3
**Technical Novelty And Significance:** 4
**Empirical Novelty And Significance:** 4
**Recommendation:** 6

**Clarity, Quality, Novelty And Reproducibility:**

The paper is well-written and generally easy/enjoyable to read. The authors do a good job spending time explaining more nuanced concepts as well. The results in the paper are high quality. The precise method in this paper I believe is original, though as the authors do mention, other non-standard clipping methods have been considered in the past.

**Strength And Weaknesses:**

I think the main strength of the paper is that it suggests a simple method for hyperparameter reduction backed by theoretical and empirical exploration that also has both impressive theoretical and empirical performance. The authors do a good job using the explorations in the first few pages of the paper to motivate the eventual automatic clipping method they arrive at in the paper. The results in the paper feel very complete and comprehensive; there are theoretical converge results, a wide range of empirical evaluations on benchmarks, and discussions on ease of implementation into existing codebases including code snippets, as well as lengthy discussions on each. Despite the main goal of the paper being to reduce the number of hyperparameters, the authors are also able to achieve empirical increases in accuracy using their method.

I think the main weakness of the paper is that it is unclear from reading the paper just how much of an improvement in terms of hyperparameter reduction this paper gives over past approaches; in particular, I felt that Figure 14 is one of the most important parts of the paper (giving examples that quantify the effect on performance of varying gamma), and should have been discussed in more detail especially in the body of the paper. In particular, the authors state that "deep learning optimizers are insensitive to the choice of gamma", but in Figure 14, the effect of gamma on accuracy and the effect of clipping norm on accuracy seem to be within a small constant factor of each other (e.g., the maximum difference b/t two clipping norms that differ by 10 seems to be < 1% absolute accuracy, and for some settings two gammas that differ by 10 can differ in accuracy by half a percent). It seems then arguable in terms of hyperparameter reduction if AUTO-S with gamma = 0.01 is substantially improving over e.g., just fixing the clip norm to be 0.01 in DP-SGD. One could reasonably argue that this constant factor is substantial in practice, but this topic is somewhat swept over by the quoted statement which to me implies the difference is much more drastic.

That being said, I think since (i) as mentioned above, gamma has less of an effect than the clipping norm and one can argue the decrease of this effect is substantial (ii) the authors also include comparisons to AUTO-V (which truly has strictly fewer hyperparameters to tune) and still see comparable performance from AUTO-V, this is a minor weakness. Furthermore, I think it's easily fixable by just adding a few sentences to the main body of the paper.

Nitpick: In various places references probably should go in parentheses, i.e. (X et al. 2022) instead of X et al. (2022) for readability, please fix.

**Summary Of The Paper:**

The paper suggests two new methods for doing clipping in DP optimization automatically, with the goal of avoiding the need to tune the clipping norm while minimally reducing (or perhaps even improving) model accuracy. The authors propose two methods. AUTO-V and AUTO-S. AUTO-V effectively rescales every gradient norm to 1, but runs into a lazy-region issue where the mean clipped gradient can be very close to even if the mean unclipped gradient are not. AUTO-S remedies this by instead multiplying each gradient g by 1/(||g|| + gamma), where gamma is a small positive constant, which gives large gradients slightly more magnitude than smaller gradients and avoids the lazy region issue. The authors give a survey of clipping thresholds in past work which motivates these clipping methods: Most past work does well when the clipping threshold is sufficiently small such that most gradients are clipped. If all gradients were clipped, then the gradient update using standard clipping can combine the learning rate and clipping norm hyperparameters, and reduces to what AUTO-V does anyway.

The authors show a theoretical convergence result for AUTO-S that applies to non-convex smooth functions. This analysis also shows the necessity of using AUTO-S over AUTO-V. Noticeably, the convergence proof has the same asymptotic dependence on T as that of non-private SGD.

The authors also run experiments using AUTO-S clipping to train image classification models, sentence classification models, and text generation models with DP. For image classification models, using the same setups as past work but replacing standard clipping with AUTO-S, the mean accuracy increases across a multitude of tasks. For sentence classification, AUTO-S improves upon the previous work of Li et al. for almost all settings, using the same hyperparameters and setup. For text generation, AUTO-S has the highest performance compared to past methods.

**Summary Of The Review:**

I recommend accepting the paper. Hyperparameter reduction is an important problem in practice, and the authors have done a very good job motivating the method they use to address this problem and making their results as well-studied and suitable for use in practice as existing methods. So I believe the paper can have high impact in practice. The empirical results in the paper are also fairly impressive, improving over the SOTA in performance despite that not being the main focus of the work necessarily. The aforementioned weakness in the paper I believe is easily fixed before the camera-ready version as well.

---

> ### Author Response · Authors · 2022-11-15
> **Thank you for liking this paper!**
>
> We thank the reviewer for the comment! We would appreciate it if the reviewer can chime in to support this work during the reviewers' discussion.
>
> **Comment:** I think the main weakness of the paper is that it is unclear from reading the paper just how much of an improvement in terms of hyperparameter reduction this paper gives over past approaches; in particular, I felt that Figure 14 is one of the most important parts of the paper (giving examples that quantify the effect on performance of varying gamma), and should have been discussed in more detail especially in the body of the paper.
>
> **Response:** We thank the reviewer for liking the paper! We will add next to Figure 1 that, because our clipping eliminates the clipping norm $R$, we may save 5 times of tuning efforts in Figure 1. We agree that the robustness of $\gamma$ is exciting and we will find space to discuss in the main text on the camera-ready version.
>
> **Comment:** In particular, the authors state that "deep learning optimizers are insensitive to the choice of gamma", but in Figure 14, the effect of gamma on accuracy and the effect of clipping norm on accuracy seem to be within a small constant factor of each other (e.g., the maximum difference b/t two clipping norms that differ by 10 seems to be < 1\% absolute accuracy, and for some settings two gammas that differ by 10 can differ in accuracy by half a percent). It seems then arguable in terms of hyperparameter reduction if AUTO-S with gamma = 0.01 is substantially improving over e.g., just fixing the clip norm to be 0.01 in DP-SGD. One could reasonably argue that this constant factor is substantial in practice, but this topic is somewhat swept over by the quoted statement which to me implies the difference is much more drastic.
>
> **Response:** We agree that AUTO-S is not substantially improving over DP-SGD with $R=0.01$. However, our core argument is the benefit of not tuning $R$ (thus saving the hyperparameter tuning effort and the privacy risk of tuning), and we are satisfied as long as the accuracy is similar.
>
> **Comment:** That being said, I think since (i) as mentioned above, gamma has less of an effect than the clipping norm and one can argue the decrease of this effect is substantial (ii) the authors also include comparisons to AUTO-V (which truly has strictly fewer hyperparameters to tune) and still see comparable performance from AUTO-V, this is a minor weakness. Furthermore, I think it's easily fixable by just adding a few sentences to the main body of the paper.
>
> **Response:** We thank the reviewer for the suggestion and we will add more sentences to the main body in the final version.
>
> **Comment:** Nitpick: In various places references probably should go in parentheses, i.e. (X et al. 2022) instead of X et al. (2022) for readability, please fix.
>
> **Response:** We have done a full sweep and corrected this!

---

### Official Review · Reviewer_LbVy · 2022-10-26

**Confidence:** 4
**Correctness:** 3
**Technical Novelty And Significance:** 2
**Empirical Novelty And Significance:** 2
**Recommendation:** 5

**Clarity, Quality, Novelty And Reproducibility:**

I found the presentation of ideas in the paper more convouted than it perhaps needs to be. The theorem statements about equivalence with fixed $R$ and different step sizes (Thm1 and Thm2) are simple observations, but their presentation seems to hint that there is some deep going on. I hope that the authors can try to simplify the presentation.
The key idea of using normalized updates is simple and novel only in the specific context of DP optimization. The quality hinges o empirical improvements over prior works; however, comparison with other adaptive clipping methods is missing and so it is not clear if these results are enough to ascertain if this method would work (significantly) better than existing counter-parts in general.

**Strength And Weaknesses:**

Strengths
1. Clipping of gradients is an integral constituent to the practice of modern differentially private machine learning. Moreover, as the authors have pointed out, the downstream performance is usually very sensitive to the choice of the clipping threshold. This has resulted in many works on how to *automatically* set the clipping threshold without tuning it explicitly. The authors of this paper contribute to this practically important area of research.

2. Extensive experiments:
The main goal of the paper is to obtain empirical improvement over prior works.
The authors propose two variants of their clipping technique, AUTO-V and AUTO-S, and perform extensive experiments on various benchmark tasks. The experimental results are promising and demonstrate better performance than compared techniques.

Weaknesses

The key idea of the paper is very simple but in my view, its presentation feels rather obfuscated.
Basically, the trick is normalize all gradients  (AUTO-V) or normalize it with an additive scalar in the denominator (which is standard in implementations of adaptive gradient methods like ADAM). This normalization explicitly controls the sensitivity, and so privacy follows. If the per-sample gradients have different norms, then this method scales all of them to have the same norm, so intuitively it seems that we would lose some information, however the authors show that it still works well in practice.


**Theoretical results**:
The presentation of the theoretical results is convoluted for some reason -- I would have expected that the right hand side in Thm 4 and 5 are stated as an explicit function of problem parameters; instead they are written as a function F which is not defined in the main text. The authors say that has the same rate as SGD (which is true with respect to its dependence on $T$), but I am not sure if the dependence on other terms is same as SGD. This should be clarified.
Also, I am not sure about the remarks following Thm5.
Aren't all these observations could be made from any existing convergence result of this form (say DP-SGD)? So, why are they interesting in the context of the methods proposed? What am I missing here?

**Connections to optimisation methods with normalized updates**: (Variants of) Normalized updates have been used in optimization; for instance, in parameter-free methods [CM20], and variance-reduced methods [FJLZ18] as well as in empirical works (see references in [CM20]).
However, I did not find any discussion on this in the paper; I hope the authors add some to give context to the proposed approach.
Also, is the theoretical result and analysis significantly different from prior works?

**Empirical Comparison with other adaptive clipping methods**:
The authors do not do any empirical comparison with prior adaptive clipping methods such as Andrew et al, Pichapati et al. The authors rightfully pointed out these introduce new (but arguably robust) hyper-parameters. However, they often come with default settings of these hyperparameters (such as the quanitile). So, why not compare using these default values of hyperparameters of all the methods while keeping the other hyperpameters (like learning rate) fixed?

[CM2020]Momentum Improves Normalized SGD
[FJLZ18]:SPIDER: Near-Optimal Non-Convex Optimization via Stochastic Path Integrated Differential Estimator

**Summary Of The Paper:**

The paper considers the problem of choosing the clipping threshold in DP-SGD routines and proposes two related modifications to the original fixed clipping technique of Abadi et al. They then show that the DP-SGD method with the proposed clipping has desirable theoretical properties in terms of convergence to stationary points. Importantly, they conduct extensive experiments to show that their method compares favourably to other schemes in practice.

**Summary Of The Review:**

The paper tackles a practically important problem and present promising empirical results. It is also interesting that it is not a heurisitc but comes with some theoretical guarantees. However, the key ideas behind it are rather simple and a comparison with competing methods is currently missing.

---

> ### Author Response · Authors · 2022-11-15
> **Thank you and response to comments**
>
> We hope our discussion can clear the confusion on Remark 5.4 and our unique contribution in per-sample normalization. Please consider raise the score if you are convinced. We are happy to discuss more if needed.
>
> **Comment:** If the per-sample gradients have different norms, then this method scales all of them to have the same norm, so intuitively it seems that we would lose some information, however the authors show that it still works well in practice.
>
> **Response:** We thank the reviewer for this comment. It is surprising that in almost all previous works, small clipping threshold $R$ leads to the best accuracy, i.e. all per-sample gradients are scaled to the same norm. This is exactly our motivation to introduce AUTO-S with stability constant $\gamma$, such that it is the only clipping function that preserves the monotonicity of gradient norm, as shown in Figure 4.
>
> **Comment:** Theoretical results: The presentation of the theoretical results is convoluted for some reason -- I would have expected that the right hand side in Thm 4 and 5 are stated as an explicit function of problem parameters; instead they are written as a function F which is not defined in the main text. The authors say that has the same rate as SGD (which is true with respect to its dependence on ), but I am not sure if the dependence on other terms is same as SGD. This should be clarified.
>
> **Response:** We thank the reviewer for this suggestion. We will clarify the dependence on other hyperparameters like privacy budget in the camera-ready version. For example, we state the convergence of non-DP SGD (under the same assumption as DP-SGD) in Appendix D Thm 9, that the model size is not affecting the convergence.
>
> Also, Thm 4 and 5 have to use an implicit form because the right hand side is an envelope function, which does not have a closed form (as we explained in Footnote 6).
>
> **Comment:** Also, I am not sure about the remarks following Thm5. Aren't all these observations could be made from any existing convergence result of this form (say DP-SGD)? So, why are they interesting in the context of the methods proposed? What am I missing here?
>
> **Response:** Our work is the first to apply a non-zero stability constant $\gamma$ in DP optimizers. Therefore there is no existing convergence result that characterizes the effect of $\gamma$. We show that $\gamma$ has to be non-zero for the gradient norm on the left hand side to vanish; otherwise, the convergence guarantee is larger than the inherent gradient noise variance $\xi$.
>
> **Comment:** Connections to optimisation methods with normalized updates: (Variants of) Normalized updates have been used in optimization; for instance, in parameter-free methods [CM20], and variance-reduced methods [FJLZ18] as well as in empirical works (see references in [CM20]). However, I did not find any discussion on this in the paper; I hope the authors add some to give context to the proposed approach. Also, is the theoretical result and analysis significantly different from prior works?
>
> **Response:** We are happy to add the discussion to the camera-ready version! In fact, these normalized SGD are fundamentally different to our automatic DP-SGD (or DP-Adam), in the same way that standard gradient clipped SGD is different to Abadi's DP-SGD. The main difference lies in the challenge of analyzing per-sample normalization (which is biased) and the batch-gradient normalization (which is unbiased in the direction):
>
> $\sum_i g_i/||\sum_i g_i||$ is parallel (in the same direction) to the non-normalized gradient $\sum_i g_i$; however, per-sample clipping $\sum_i (g_i/||g_i||)$ has bias that is difficult to analyze.
>
> Therefore, our theoretical result and analysis is **completely different** from prior works, as we show that even with gradient bias, DP-SGD can still converge.
>
> **Comment:** Empirical Comparison with other adaptive clipping methods: The authors do not do any empirical comparison with prior adaptive clipping methods such as Andrew et al, Pichapati et al. The authors rightfully pointed out these introduce new (but arguably robust) hyper-parameters. However, they often come with default settings of these hyperparameters (such as the quanitile). So, why not compare using these default values of hyperparameters of all the methods while keeping the other hyperpameters (like learning rate) fixed?
>
> **Response:** We agree with the reviewer that adaptive clipping could potentially be interesting to compare to. We don't experiment with them because (1) Adaptive clipping is kind of limited or ad-hoc in the setting: why use 90\% quantile not 91\%? what fraction of privacy budget should be used to privately release the quantile (if there is no public data like our experiments)? While we can use the default settings for comparison, such approach may leads to unfair baselines; (2) Adaptive clipping has not been applied to large models with hundreds of millions of parameters (at least without public codebase).

---

### Official Review · Reviewer_adTx · 2022-10-26

**Confidence:** 4
**Correctness:** 3
**Technical Novelty And Significance:** 2
**Empirical Novelty And Significance:** 2
**Recommendation:** 3

**Clarity, Quality, Novelty And Reproducibility:**

Clarity: the paper is relatively clear

Quality: the quality of the work is ok, including detailed experiments.

Novelty: the novelty is limited, since the two clipping functions are very similar, and prior work has established that the standard clipping function is also very easy to tune.

Reproducibility: the work is reproducible

**Strength And Weaknesses:**

Strengths:

1) The proposed scheme is a natural alternative clipping scheme.
2) The authors provide extensive experiments to confirm that it achieves similar performance to standard clipping

Weaknesses:

1) The core claim of the paper is that the clipping parameter R is difficult to tune, however a number of prior works have shown that this arises only because the choice of clipping norm changes the scale of the update. One can easily fix this by redefining $Clip_R(g) = min(1/|g|, 1/R)$. As De et al. show, after this transformation a wide range of small but finite values of R perform similarly well, and it is standard practice in recent works to simply set R=1.

2) After this transformation, it is clear that the two clipping schemes are almost identical. Essentially the authors propose a smoothed version of the standard clipping function.



**Summary Of The Paper:**

The authors propose to replace the standard clipping function in DP-SGD, $Clip_R(g) = min(R/|g|, 1)$ with the alternative clipping function $Clip_{\gamma}(g) = 1/(|g| + \gamma)$. They argue that this alternative clipping scheme is easier to tune, and matches or exceeds the performance of the standard scheme on a range of datasets.

**Summary Of The Review:**

The paper relies on the claim that the clipping threshold in DP-SGD is difficult to tune, however recent works have shown that this is easy in practice if one simply re-scales the clipped gradient by the clipping threshold. Unfortunately I therefore feel the contribution here is simply too small for a top-tier conference.

---

> ### Author Response · Authors · 2022-11-15
> **Thank you and emphasis on overlooked contributions**
>
> We thank the reviewer for the comments and would like to bring some over-looked contributions to your attention. Please consider raising the score if you are convinced.
>
> **Comment:** The core claim of the paper is that the clipping parameter R is difficult to tune, however a number of prior works have shown that this arises only because the choice of clipping norm changes the scale of the update. One can easily fix this by redefining . As De et al. show, after this transformation a wide range of small but finite values of R perform similarly well, and it is standard practice in recent works to simply set R=1.
>
> **Response:** Thank you for initiating the discussion on concurrent works. We notice that the reviewer only recognizes one out of many of our claims. We provide a list of major contributions that are **unique** in this work, but not in De et al.:
>
> 1. Our clipping **rigorously** removes the clipping threshold $R$, where previous works only **approximately** remove the need to tune $R$. In previous works, the independence on $R$ only holds true for a range of small $R$, but one still needs tuning efforts (maybe inexpensive if one is lucky) to find this range. Also, the re-parameterization trick by De et al. only explains DP-SGD and won't explain the more powerful DP-Adam. We quote in their paper "This is a re-parameterization of Equation (2) in which the learning rate $\eta$ absorbs a factor of $C$." As we explicitly point out in Section 4.1 and 4.2, DP-Adam's learning rate *DOES NOT* absorb the clipping threshold. Instead, the clipping threshold is absorbed by its adaptive denominator. Therefore, DP-SGD and DP-Adam have totally different patterns as shown in Figure 1.
>
> 2. We are excited about our **theoretical analysis**, which is the first of its kind in the DP literature. The reviewer seems to overlook this core contribution, which not only reveals the significance of the stability constant $\gamma$ but also provides practical guidance on training hyperparameters, e.g. batch size, number of epochs, ... (see bullet points below Theorem 5).
>
> 3. Our experiments are extensive, especially on the **language tasks**, whereas De et al. only works with image tasks (which we additionally have CelebA that contains real human faces). Theses experiments have empirically confirmed the significance of our clipping method.
>
> We hope the reviewer can re-evaluate our work based on all contributions, not just on the form of our method.
>
> **Comment:** After this transformation, it is clear that the two clipping schemes are almost identical. Essentially the authors propose a smoothed version of the standard clipping function.
>
> **Response:** We agree that our clipping and De et al.'s re-parameterized Abadi's clipping is similar. However, the **difference is subtle but important**: the stability constant $\gamma$, which does not exist in De et al. or any previous work. In theory (our Theorem 4), we show that non-zero $\gamma$ is key to better convergence. In experiments (see Table 1,2,3,4), we demonstrate that non-zero $\gamma$ has consistently higher accuracy. We provide deep insight in Figure 4: only AUTO-S is monotone in the per-sample gradient norm.

---

> > ### Comment · Reviewer_adTx · 2022-11-22
> > **Clarifying my concerns**
> >
> > To clarify my concerns, if you rename $\gamma$ to $R$, then the proposed clipping takes the form $Clip_R(g) = 1/(|g| + R)$, while rescaled Abadi's clipping takes the form $Clip_R(g) = min(1/|g|, 1/R)$.
> >
> > If you plot these two clipping functions against each other across a wide range of values of $|g|$, you will see that they are almost identical, both taking exactly the same value when $|g| \gg R$ or $|g| \ll R$, while the proposed method is simply smoother near $|g| = R$. I therefore do not think that this method can be described as "automatic" or as removing $R$, since the new hyper-parameter $\gamma$ plays an near-identical role to the previous hyper-parameter $R$. It will be similarly difficult/easy to tune and it will take similar optimal values.
> >
> > I agree with the authors that it makes sense to consider smoothed versions of the clipping, since it is plausible that a smoothed scheme might achieve slightly better performance in practice. I think there is the potential for an interesting empirical paper comparing these two clipping schemes across a range of tasks, but I don't think the paper should be accepted with its current motivation. I'd encourage the authors to modify the description of the proposed method for a re-submission.

---

> > > ### Author Response · Authors · 2022-11-22
> > > **Not to confuse function similarity and effect**
> > >
> > > We thank the reviewer for continued discussion! A short response would be:
> > >
> > > 1. Yes, **our clipping and Abadi's clipping are similar when $||g_i||$ is large, but that doesn't mean the effects of two clippings are similar.** You have to take into account the distribution of $||g_i||$: many per-sample gradient norms are not large, thus the effects are different as we demonstrate in experiments.
> > >
> > > We would like to emphasize that all clipping functions $C(||g_i||)$ needs to satisfy $C<R/||g_i||$ (if the added noise is $\sigma R N(0,I)$; or $C<1/||g_i||$ if the added noise is $\sigma N(0,I)$), in order to have $||g_i\times C_i||<R$. Therefore the Abadi's clipping and our automatic clipping will be similar when $||g_i||$ is large enough.
> > >
> > > In fact, you don't need to consider the rescaled Abadi's clipping and please check our Figure 4 (where both automatic and Abadi's clippings are not rescaled). The $\gamma=0.01$ and Abadi's clipping are similar for $||g_i||>1$. But interesting things happen when $||g_i||<R$ in SOTA experiment settings. For example, in the 6 plots of Figure 12 (appendix), the percentage of not-clipped gradients (i.e. $||g_i||<R$) are 40-70%. Our clipping has very different effect on these per-sample gradients than Abadi's.
> > >
> > > 2. We emphasize that **our clipping is automatic** in the following two manners.
> > >
> > > First, AUTO-V is completely free of $R$ (hence definitely automatic, and should serve as baseline for future works) and AUTO-S is very robust against $\gamma$ (see Figure 14 appendix). This is why we always **fix** $\gamma=0.01$ across experiments. Notice that Abadi's clipping is not robust against $R$ as shown in Figure 1.
> > >
> > > Second, $\gamma$ does not "plays a near-identical role to the previous hyper-parameter R". Besides the robustness issue mentioned above, we would also like to point out that AUTO-V is actually the version of Abadi's clipping when $R$ is infinitely small, thus $\min(R/|g_i|,1)$ is always $R/|g_i|$. Because AUTO-S is a smoothed version, it is also similar to infinitely small $R$ Abadi's clipping, not to arbitrary $R$ (i.e. for all $\gamma$, we always look at infinitely small $R$; this $\gamma$ to $R$ mapping is not one-to-one). This has been argued in Section 3.1 second paragraph. In other words, we advocate infinitely small $R$ (a single R, **not to tune**), which turns out to be normalization AUTO-V, which can further be smoothed to AUTO-S (a single $\gamma=0.01$, **not to tune**).

---

### Decision · Program_Chairs · 2023-01-20

**Decision:**

Reject

**Justification For Why Not Higher Score:**

Limited advantage and novelty versus prior methods.

**Justification For Why Not Lower Score:**

N/A

**Metareview: Summary, Strengths And Weaknesses:**

This work largely focuses on removing the clipping norm hyperparameter, but replacing it with another hyperparameter which the authors show that the optimizer is relatively insensitive to. The modification is fairly similar to a previous reparameterization made by De et al. Some reviewers were also critical of the writing. One back-and-forth between a reviewer and the authors also seemed to indicate that it would get comparable performance to simply fix the clipping norm, which obviates the need for the modification in this paper. No reviewer was willing to champion this paper.